# Effects of Zn Exposure on *Populus simonii* Seedling Growth and Its Resistance to Leaf Rust

**Lulu Gao** [1,2,†], **Aoying Zhang** [1,2,†] **and Shanchun Yan** [1,2,*]

1   School of Forestry, Northeast Forestry University, Harbin 150040, China; aoyingzhang@nefu.edu.cn (A.Z.)
2   Key Laboratory of Sustainable Forest Ecosystem Management-Ministry of Education,
    Northeast Forestry University, Harbin 150040, China
*   Correspondence: yanshanchun@nefu.edu.cn
†   These authors contributed equally to this work.

**Abstract:** Heavy metals are serious pollutants that affect the growth and disease resistance of woody plants. Herein, the enrichment characteristics of an essential element (Zn) in *Populus simonii* seedlings, as well as the effects of Zn stress on seedling growth and resistance to leaf rust, were investigated. Zn contents in roots, stems, and leaves of treatment groups were significantly increased. Zn stress at all concentrations significantly inhibited the biomass of seedlings. Under the low, middle, and high Zn treatments, compared with the control group, CAT activity significantly decreased by 36%, 21%, and 14%; SOD activity significantly decreased by 13%, 8%, and 5%; PPO activity significantly decreased by 27%, 31%, and 47%; TI activity significantly decreased by 48%, 55%, and 61%, and CI activity significantly decreased by 10%, 20%, and 14%, respectively. In the natural environment, we accidentally discovered that *P. simonii* was infected with leaf rust, and Zn stress significantly increased the rust disease index. The disease index correlated positively with Zn content in leaves and negatively with the chemical defense indexes. Taken together, Zn stress causes a strong growth toxicity in *P. simonii* seedlings, and the reduced chemical defense under Zn stress increases the susceptibility of seedlings to leaf rust.

**Keywords:** essential element; *Populus simonii*; growth; chemical defense; leaf rust

## 1. Introduction

Soil is an important part of the natural ecosystem. With the rapid development of industry, agriculture, and transportation, abundant heavy metals including Cd, Zn, Pb, Cr, etc., flow into the soil environment [1,2]. When heavy metals in the soil exceed a certain level, this can cause a series of ecotoxicological effects, such as soil degradation and ecosystem damage [3,4]. Currently, soil heavy metal pollution has become a global environmental problem. In China, heavy metals such as Cd, Pb, As, and Zn are the main causes of soil pollution, and about one-sixth of agricultural land is polluted by heavy metals [5,6]. The European Union has ~3.5 million polluted sites, of which ~50 sites have reached high pollution levels [7].

As the substrate of plant growth, soil can exchange materials, energy, and information with plants and provide the necessary energy and materials for plant growth [8]. Heavy metals in the soil are absorbed by plant roots during the process of material exchange between the soil and plants and are finally transported to various parts of the plants [9]. An accumulation of heavy metal ions can induce plants to increase the production of reactive oxygen species (ROS), with negative impacts on plant development [10,11]. For example, Parmar et al. (2013) found that Cd can block chlorophyll synthesis by replacing $Mg^{2+}$ in chlorophyll [12]. Milone et al. (2003) found that Cd can significantly inhibit the photosynthetic efficiency of plants, thereby impairing plant growth [13]. To counter the toxicity of heavy metals, plants usually employ effective detoxification mechanisms such as avoidance, subcellular distribution, and detoxification metabolism [14]. Avoidance means

that plants precipitate heavy metal ions through root adsorption and discharge to the outside environment [15]. Subcellular distribution refers to the conversion of the absorbed heavy metals into low or non-bioactive forms, which are then stored in inactive cellular components such as vacuoles and cell walls [16,17]. Finally, the detoxification metabolism means that plants scavenge ROS and reduce oxidative damage by activating the antioxidant defense system in vivo [18,19]. The antioxidant defense system is made up of antioxidant enzymes, such as catalase (CAT), peroxidase (POD), and superoxide dismutase (SOD), and non-enzymatic antioxidants, including glutathione and ascorbic acid [20].

As proposed by the induced defense hypothesis, stress from heavy metal exposure induces plants to produce phytochemical defense-related substances [21]. These substances involve defense enzymes, such as phenylalanine ammonia-lyase (PAL) and polyphenol oxidase (PPO), and protease inhibitors, including trypsin inhibitors (TIs) and chymotrypsin inhibitors (CIs), which can increase the plant's defense capability [22]. Additionally, the elemental defense hypothesis proposes that heavy metal enrichment in plants can also improve the resistance of plants to pathogenic microorganisms or herbivorous insects [23,24]. For example, Martos et al. (2016) found increased resistance to *Alternaria alternata* in Zn-enriched *Arabidopsis thaliana* [24]. However, the defense mechanism was dependent on the heavy metal species, its concentration, and on the host plant species. Dai et al. (2020) found higher resistance to *Rhizoctonia solani* in *Eupatorium adenophora* treated with lower rather than higher doses of Cd [25].

Zn is an essential element for plant growth that promotes the photosynthesis rate and nutrient biosynthesis [26,27]. When the environmental Zn concentration exceeds the plant tolerance threshold, it will still cause toxicity in plants [28]. *Populus simonii* is often used as a common tree species in the fields of architecture, urban greening, and ecological protection because of its strong adaptability, fast growth, and both cold and drought resistance [29]. These are the key areas where heavy metal pollution occurs, resulting in a higher exposure risk of *P. simonii* to heavy metal. With the rapid development of bioassay experiments of heavy metals, many recent studies have systematically explored the phytotoxic effects and their associated mechanisms of heavy metals. However, very few have studied the growth and physiological status of trees under the stress of essential metal elements, as well as their disease resistance ability in natural environments. Based on this, the objective of this work was to evaluate the influence of different Zn concentrations on plant growth and on the poplar leaf rust resistance of *P. simonii* seedlings.

## 2. Materials and Methods

### 2.1. Experimental Design

Three different concentrations of $ZnCl_2$ (300 mg/kg, 500 mg/kg and 700 mg/kg) solution were used to treat the soil of biennial potted *P. simonii*. After 2 months of Zn stress, the accumulation and transfer effects of Zn in various tissues of *P. simonii*, as well as the growth of *P. simonii*, were analyzed. Subsequently, the antioxidant enzymes activities, plant defense enzymes activities, and protease inhibitor content of *P. simonii* after 2 months of Zn treatment were measured. In addition, the incidence and disease index of leaf rust of *P. simonii* in the control group and each Zn treatment group were statistically analyzed under natural conditions.

### 2.2. Experimental Materials and Treatment

On 30 April 2020, biennial *P. simonii* seedlings were purchased from Harbin Flower Market (Harbin, China) and were planted in 3 L flowerpots (25 cm diameter by 23 cm height) containing 3 kg substrate in the Maoershan Experimental Forest Farm, Harbin City, Heilongjiang Province, China. Maoershan Experimental Forest Farm has a temperature of about 10–27 °C and relative humidity of about 50%–70% during May to September. One seedling was planted in each flowerpot. The substrate properties are shown in Table S1. Insect-proof nets were used to cover the space around and above the tested plants to prevent insect attacks (Figure S1). During the experiment, water was poured every 3 days and grass

was removed twice a month. After two months' growth, healthy *P. simonii* seedlings with similar growth were divided into four groups. Among them, the soils in three groups were treated with $ZnCl_2$ solution (Aladdin, China), with the treatment concentration of $Zn^{2+}$ reaching 300 mg/kg (denoted as $Zn_{300}$), 500 mg/kg (denoted as $Zn_{500}$), and 700 mg/kg (denoted as $Zn_{700}$). The other group was treated with the same amount of distilled water solution, which was recorded as the control (CK). The treatment concentration of Zn was set according to the treatment concentration in *Populus alba* × *P. berolinensis* by Jiang et al. [30] $ZnCl_2$ solution or distilled water was injected into the substrate using a syringe. Before and after Zn treatment, all the seedlings were planted in pots without transplanting. A total of 150 seedlings were planted in each group. Before that, a tray was placed under the plastic flowerpot to prevent the seedling root from penetrating underground and thereby affecting the experimental effect. Leaf samples for physiological measurements were collected on the 50th day after Zn treatment and then stored in a −80 °C ultra-low temperature refrigerator. Each group of leaf samples had 3 replicates, and each replicate consisted of all leaves in 3 seedlings. The whole plant samples were collected on the 60th day after Zn treatment.

*2.3. Determination of Heavy Metal Content in P. simonii*

The whole seedling of *P. simonii* was divided into the roots, stems, and leaves. All the samples of roots, stems, leaves, and soil were dried at 70 °C to constant weight. The sample was crushed through a 100-mesh sieve and put into a sealed polyethylene bag for storage at room temperature. For plant samples, 0.2 g samples of roots, stems, and leaves were placed in a 50 mL conical flask, and 5 mL of $HNO_3$ (68%) and 2 mL of $HClO_4$ (70%) were added and left at 25 °C for 12 h. The conical flask was then placed on a graphite digester (YXGD-I-36, Tianjin, China) at 160 °C for 10 h. After digestion was completed, the final digested liquid (about 1 mL) was cooled to room temperature. For soil samples, 0.15 g of soil was placed in a 50 mL conical flask, 5 mL of HCl solution was added into it, and the flask was placed on the graphite digester for 2 h. Then, 2.5 mL of $HNO_3$ (68%), 2.5 mL of HF (40%), and 1.5 mL of $HClO_4$ (70%) solutions were added to the flask, and digestion continued for 10 h. Subsequently, the digestion solution was cooled down to room temperature (25 °C), poured into a 25 mL volumetric flask, and the volume was made up to 25 mL with 2% $HNO_3$ solution. The Zn content in samples was determined using an atomic flame spectrophotometer (ICE3000, Waltham, MA, USA). Each group had 3 replicates, and each replicate consisted of 3 seedlings. The enrichment coefficient and transport coefficient of Zn in *P. simonii* were calculated according to the heavy metal concentrations in the soil and different parts of *P. simonii*. The enrichment coefficient indicated the ability of heavy metals in the soil to migrate into plants. The transfer coefficient represented the ability of the plant to transfer heavy metals from underground parts to aboveground parts. The coefficients were calculated based on the formula reported by Jiang et al. (2020) as follows [31]:

Enrichment coefficient = (Zn content in different parts of plants)/(Zn content in soil);

Transfer coefficient = (Zn content in aboveground part)/(Zn content in underground part).

*2.4. Determination of Growth Indexes*

Five seedlings, each one considered a biological replicate, were randomly selected for whole plant sampling. First, the plant height, ground diameter, and root length of each seedling were measured. The plant was then divided into three parts: root, stem, and leaf, and their fresh weights were measured. The roots, stems, and leaves of each seedling were dried in an oven to constant weight at 70 °C, and their dry weights were subsequently measured.

### 2.5. Determination of Antioxidant Enzyme Activity

The 0.3 g of cryopreserved leaves along with 6 mL of 50 mM sodium phosphate buffer (pH = 7.8) were ground into a homogenate in the liquid nitrogen bath. The homogenate was then centrifuged at 4 °C and 10,000 rpm for 10 min, and the supernatant was taken as the crude extract. The CAT activity in the leaves of *P. simonii* was determined using the hydrogen peroxide method [32]. CAT can decompose $H_2O_2$ into water and oxygen, and the activity of CAT can be measured according to the consumption of $H_2O_2$. The reaction system contained 2 mL sodium phosphate buffer solution (pH 7.8), 1 mL 0.1 M $H_2O_2$, and 0.05 mL crude extract. The absorbance at 240 nm was measured immediately after all the reagents of the reaction system were added, and then recorded every 1 min for 5 min. OD value of the mixture was determined by elisa reader (Shanghai Flash Spectrum Biotechnology Co., Ltd., SuPerMax 3100, Shanghai, China). The change of absorbance per gram of fresh sample per minute was 0.01 as an enzyme activity unit, which was expressed as U/GFW/min. SOD activity in *P. simonii* leaves was determined by the nitroblue tetrazolium (NBT) method, according to the principle that SOD can inhibit the reduction of NBT under light [31]. The reaction system comprised 1.5 mM phosphate buffer, 0.3 mM phosphomethionine, 0.3 mL 750 μM NBT, 0.3 mL 100 μM $Na_2$-EDTA, 0.3 mL 20 μM riboflavin, 0.05 mL crude extract, and 0.25 mL distilled water. The total volume was 3 mL. The glass finger tube was prepared. Two tubes were control tubes, and the rest were sample tubes. The crude extract was not added to the two control tubes; an equal volume of distilled water was used instead. One of control tubes was placed in darkness for 20 min, and the other control tube and all sample tubes were placed under the light intensity of 4000 Lx for 20 min. Subsequently, the absorbance of each tube at 560 nm was immediately determined by zeroing the absorbance value of the control tube with the dark treatment. OD value of the mixture was determined by elisa reader (Shanghai Flash Spectrum Biotechnology Co., Ltd., SuPerMax 3100). According to the absorption value of the control tube with the light treatment, the photochemical reduction of nitroblue tetrazolium inhibited by 50% per gram of fresh samples was taken as one unit of enzyme activity. The SOD activity was expressed as U/GFW.

### 2.6. Effects of Zn Stress on the Chemical Defense of P. simonii

The PAL activity in *P. simonii* leaves was determined by phenylalanine colorimetry. First, 0.3 g of leaves along with 5 mL of 0.1 M mercaptoethanol and 1 mL of 10% polyvinylpyrrolidone (PVP) solution were ground into a homogenate in a liquid nitrogen bath, centrifuged for 10 min at 4 °C and 10,000 rpm, and the resulting supernatant was considered as the crude extract. The reaction system contained 3.8 mL of boric acid buffer, 1 mL of 0.02 mM L-phenylalanine solution, and 0.2 mL of crude extract. The reaction was carried out in a 40 °C water bath for 60 min, and the reaction was finally terminated by adding 0.2 mL HCl. The absorbance at 290 nm was measured immediately and repeated five times. OD value of the mixture was determined by elisa reader (Shanghai Flash Spectrum Biotechnology Co., Ltd., SuPerMax 3100). The change of absorbance per minute per gram of fresh sample was taken as a unit of enzyme activity and expressed as U/GFW/min. Triplicates were measured for each group.

The PPO activity in the leaves of *P. simonii* was determined by catechol colorimetry. First, 0.3 g of cryopreserved leaves along with 6 mL of PBS solution containing 7% polyvinylpyrrolidone (PVP) were ground into a homogenate in a liquid nitrogen bath and centrifuged at 5500 rpm for 10 min at 4 °C, and the resulting supernatant was taken as the crude extract. The reaction system contained 1.5 mL of 0.1 M catechol solution, 1 mL of 0.1 M PBS buffer, and 0.5 mL of crude extract. After the reaction was completed, the absorbance at 420 nm was measured, recorded every 30 s, and repeated five times. OD value of the mixture was determined by elisa reader (Shanghai Flash Spectrum Biotechnology Co., Ltd., SuPerMax 3100). The change of absorbance per gram of fresh sample per minute was taken as an enzyme activity unit, which was expressed in U/GFW/min. Triplicates were measured for each group.

For TI activity determination, 0.3 g of cryopreserved leaves along with 6 mL of Tris-HCl buffer containing $CaCl_2$ were ground into a homogenate in a liquid nitrogen bath and centrifuged at 12,000 rpm for 10 min at 4 °C, and the resulting supernatant was considered the crude extract. Then, 60 μL of protease with known activity and 40 μL of crude extracts were reacted in the water bath at 25 °C for 60 min. The reaction system contained 2.9 mL of BAEE and 0.1 mL of the abovementioned mixture and was allowed to stand for 1 min. The absorbance at 256 nm was measured immediately and recorded every 60 s, and this was repeated five times. OD value of the mixture was determined by elisa reader (Shanghai Flash Spectrum Biotechnology Co., Ltd., SuPerMax 3100). The change of absorbance per gram of fresh sample per minute was taken as one TI activity unit, and the TI activity was expressed as U/GFW/min. Triplicates were measured for each group.

For CI activity determination, 0.3 g cryopreserved leaves along with 6 mL Tri-HCl buffer (0.05 M, pH 8.8) containing vitamin C, phenyl thiourea, polyvinyl alcohol, and $CaCl_2$ were ground into a homogenate in a liquid nitrogen bath, centrifuged at 12,000 rpm for 10 min at 4 °C. The resulting supernatant was the crude extract. Then 60 μL protease with known activity and 40 μL crude extracts reacted in the water bath at 25 °C for 60 min. The reaction system contained 2.9 mL BAEE and 0.1 mL of the abovementioned mixture and was allowed to stand for 1 min. The absorbance at 253 nm was measured immediately and recorded every 60 s, and this was repeated five times. OD value of the mixture was determined by elisa reader (Shanghai Flash Spectrum Biotechnology Co., Ltd., SuPerMax 3100). The change in absorbance per gram of fresh sample per minute was taken as a CI activity unit, and the CI activity was expressed as U/GFW/min. Triplicates were measured for each group.

### 2.7. Sensitivity Analysis of P. simonii to Leaf Rust

On 31 August 2020, under natural conditions, we found that *P. simonii* in the CK and Zn treatment groups were infected with leaf rust. Based on this, we observed the incidence and disease index of the leaf rust of *P. simonii* in the treatment and control groups. In brief, 90 *P. simonii* seedlings were selected in each group to investigate the rust infection of *P. simonii*. For each plant, the presence of at least one leaf with rust was considered as a diseased plant. According to Feng et al., (2008) [33], the incidence rate was calculated by the following formula:

Incidence = (number of diseased plants)/(total number of plants). The incidence was expressed as %. Triplicates were measured for each group.

All the leaves of diseased *P. simonii* were collected, and the disease incidence degree of all leaves in the diseased plants was counted. The statistical methods of disease grading of *P. simonii* leaves are shown in Table 1. According to Feng et al., (2008) [33], the formula for calculating the disease index of *P. simonii* is as follows:

**Table 1.** Statistical method of disease grade in the diseased seedlings.

| Severity Index | Incidence Degree |
| --- | --- |
| 0 | Disease-free leaves |
| 1 | Infected leaves with less than 1/4 of the leaf area |
| 2 | Infected leaves with 1/4–1/2 of the leaf area |
| 3 | Infected leaves with 1/2–3/4 of the leaf area |
| 4 | Infected leaves with more than 3/4 of the leaf area |

Disease index = Σ (representative value of disease grade × number of leaves of disease grade)/(representative value of maximum incidence grade × total number of leaves investigated) × 100%. Triplicates were measured for each group.

### 2.8. Statistical Analysis

After testing the variance homogeneity and normal distribution, the significance of data among control/treatment groups was analyzed by one-way ANOVA with a multiple

LSD comparison at α = 0.05. The Pearson correlation analysis was carried out to determine the correlation between incidence rate (or disease index) and physiological parameters using the Lc-bio Technologies Cloud Platform (https://www.omicstudio.cn/index, accessed on 1 October 2022).

## 3. Results

### 3.1. Enrichment and Transport Ability of Zn in P. simonii

In soil, the Zn content of the $Zn_{300}$ treatment group was not significantly different from that of the CK, but the contents of the $Zn_{500}$ and $Zn_{700}$ groups were significantly higher than that of the CK (Figure 1A). The Zn content of the different treatment groups in the roots, stems, and leaves of *P. simonii* was positively correlated with the treatment concentration ($r = 0.986$ and $p < 0.01$ for roots; $r = 0.932$ and $p < 0.01$ for stems; $r = 0.972$ and $p < 0.01$ for leaves) (Figure 1B–D). In the $Zn_{300}$, $Zn_{500}$, and $Zn_{700}$ treatment groups, compared with the control group, Zn content in the root significantly increased by 129%, 198%, and 349%, Zn content in the stem significantly increased by 30%, 178%, and 379%, and Zn content in the leaf significantly increased by 83%, 254%, and 401%, respectively. However, the enrichment coefficient of *P. simonii* roots in all Zn treatment groups was significantly higher than that of the CK and was negatively correlated with the Zn treatment concentration ($r = -0.887$ and $p < 0.01$). The enrichment coefficients of the stems and leaves of *P. simonii* were significantly higher than that of the control but were specific to the Zn concentration (Table S2). The transport coefficient of the $Zn_{300}$ treatment group was significantly lower than that of the control group, while those of the $Zn_{500}$ and $Zn_{700}$ treatment groups were significantly higher than that of the CK group, while the transport coefficient of the $Zn_{700}$ treatment group was significantly higher than that of the $Zn_{500}$ treatment group (Table S2).

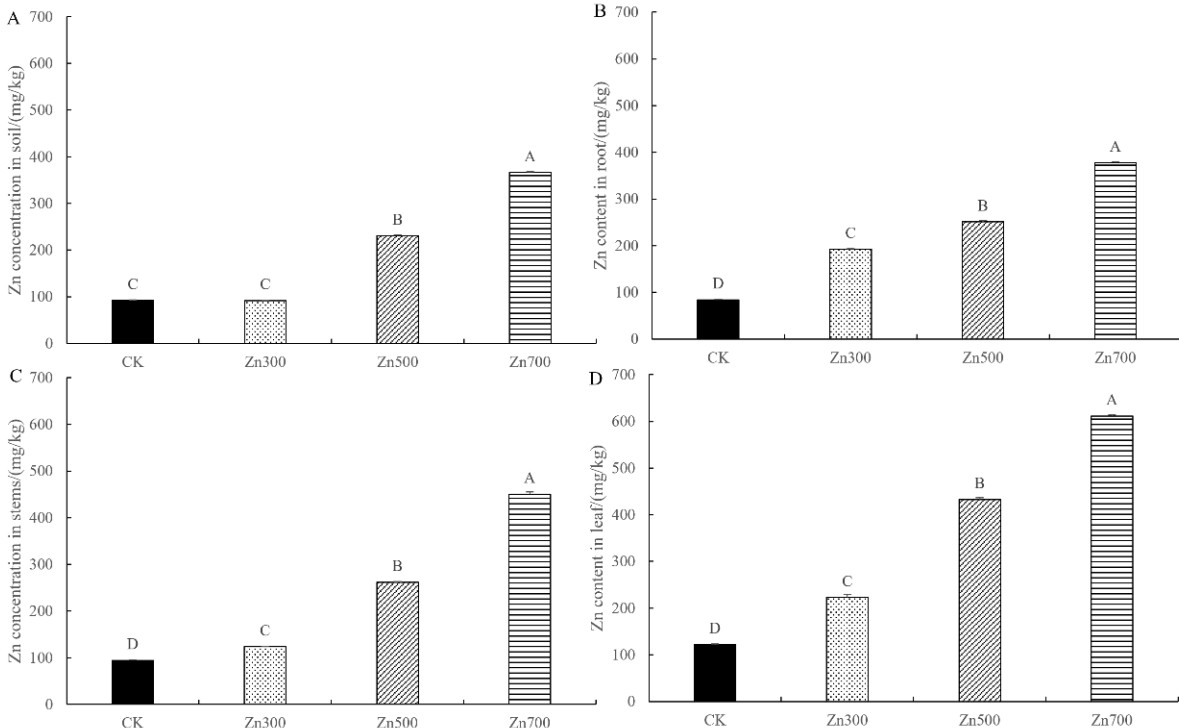

**Figure 1.** Zn enrichment in the soil (**A**), as well as the roots (**B**), stems (**C**), and leaves (**D**) of *P. simonii*. The values shown in the figure are the average value ± SD (n = 3). Different letters indicate that there are significant differences among groups (one-way ANOVA, $p < 0.05$).

### 3.2. Effects of Zn Stress on the Growth of P. simonii

To evaluate the toxic effects of different Zn treatments on woody plants, we measured the growth of *P. simonii* seedlings in the CK and different Zn treatment groups. The

plant height, fresh root weight, and dry root weight of the $Zn_{300}$ treatment group were significantly lower than those of the control group, but the other growth parameters did not differ significantly. There was no significant difference between the $Zn_{500}$ treatment and CK groups, with only the fresh and dry root weights being significantly lower. Furthermore, all the indexes of growth and development in the $Zn_{700}$ treatment group were significantly lower than those in the CK group (Table 2).

**Table 2.** Effects of Zn treatment on the growth and biomass of *P. simonii*.

| Groups | Plant Height (cm) | Root Fresh Weight (g) | Root Dry Weight (g) | Stem Fresh Weight (g) | Stem Dry Weight (g) | Leaf Fresh Weight (g) | Leaf Dry Weight (g) |
|---|---|---|---|---|---|---|---|
| CK | 134.5 ± 5.54A | 10.18 ± 0.55A | 7.12 ± 0.24A | 23.79 ± 4.3A | 14.45 ± 2.76A | 30.77 ± 6.05A | 14.45 ± 2.76A |
| $Zn_{300}$ | 95.9 ± 4.54B | 7.19 ± 0.12C | 4.9 ± 0.23B | 20.25 ± 3.66AB | 10.9 ± 1.42AB | 25.78 ± 4.06AB | 10.9 ± 1.42AB |
| $Zn_{500}$ | 125.4 ± 7.14A | 8.8 ± 0.57B | 5.47 ± 0.47B | 19.35 ± 2.65AB | 11.19 ± 1.47AB | 25.44 ± 2.89AB | 11.19 ± 1.47AB |
| $Zn_{700}$ | 83.9 ± 1.15B | 6.09 ± 0.27C | 5.32 ± 0.43B | 12.27 ± 0.5B | 5.96 ± 0.21B | 14.59 ± 0.59B | 5.96 ± 0.21B |

The values shown in the figure are the average value ± SD (n = 5). Different letters indicate that there are significant differences among groups (one-way ANOVA, $p < 0.05$).

### 3.3. Response of Antioxidant Defense in P. simonii to Zn Stress

The response trend of the antioxidant defense system of *P. simonii* seedlings to Zn stress was evaluated (Figure 2). In the $Zn_{300}$, $Zn_{500}$, and $Zn_{700}$ treatment groups, compared with the control group, CAT activity significantly decreased by 36%, 21%, and 14%, and SOD activity significantly decreased by 13%, 8%, and 5%, respectively. In the treatment group, the CAT and SOD activities were positively correlated with the Zn treatment concentration ($r = 0.961$ and $p < 0.01$ for CAT; $r = 0.955$ and $p < 0.01$ for SOD), with the high Zn treatment group having significantly higher CAT and SOD activities than the low or medium Zn treatment groups.

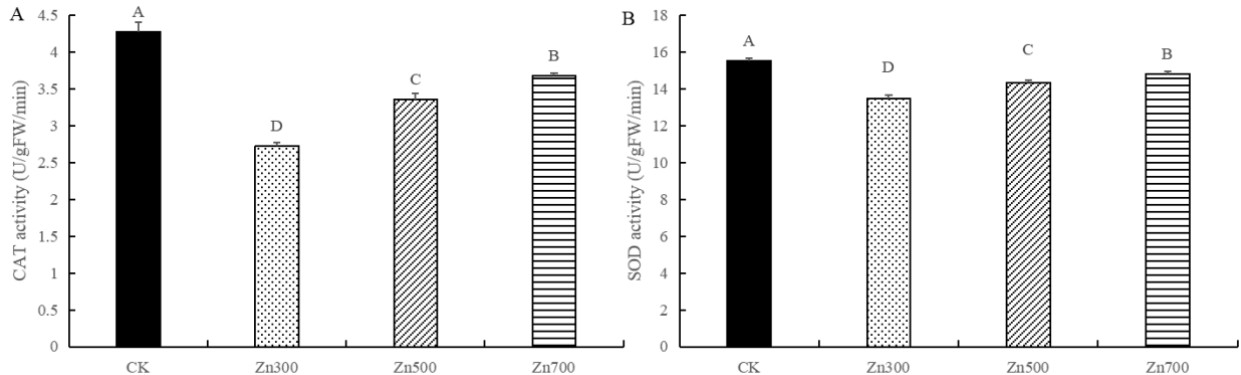

**Figure 2.** Effects of Zn treatment on CAT (**A**) and SOD (**B**) activities of *P. simonii*. The values shown in the figure are the average value ± SD (n = 3). Different letters indicate that there are significant differences among groups (one-way ANOVA, $p < 0.05$).

### 3.4. Effects of Zn Stress on Chemical Defense of P. simonii

We determined the activities of defense enzymes and protease inhibitors in the leaves of *P. simonii* to determine the chemical defense response trend to Zn stress. There was no significant difference in the PAL activity between the treatment and the control groups, but the PAL activity of the $Zn_{700}$ treatment group was significantly lower than those of the $Zn_{300}$ and $Zn_{500}$ treatment groups (Figure 3A). Each treatment group had significantly lower PPO activity than the CK group, with the $Zn_{700}$ group being lower than the $Zn_{300}$ and $Zn_{500}$ groups. In the $Zn_{300}$, $Zn_{500}$, and $Zn_{700}$ treatment groups, compared with the control group, PPO activity decreased by 27%, 31%, and 47%, respectively (Figure 3B). For TI activity, each treatment group had significantly lower activity than the CK group, with the $Zn_{700}$ group being significantly lower than the $Zn_{300}$ treatment group. In the $Zn_{300}$, $Zn_{500}$, and $Zn_{700}$ treatment groups, compared with the control group, the TI activity decreased

by 48%, 55%, and 61%, respectively (Figure 3C). For CI activity, the treatment groups had significantly lower activity than the control group, the $Zn_{700}$ group was significantly lower than the $Zn_{300}$ group, and the $Zn_{500}$ treatment group was significantly lower than the $Zn_{300}$ and $Zn_{700}$ treatment groups (Figure 3D).

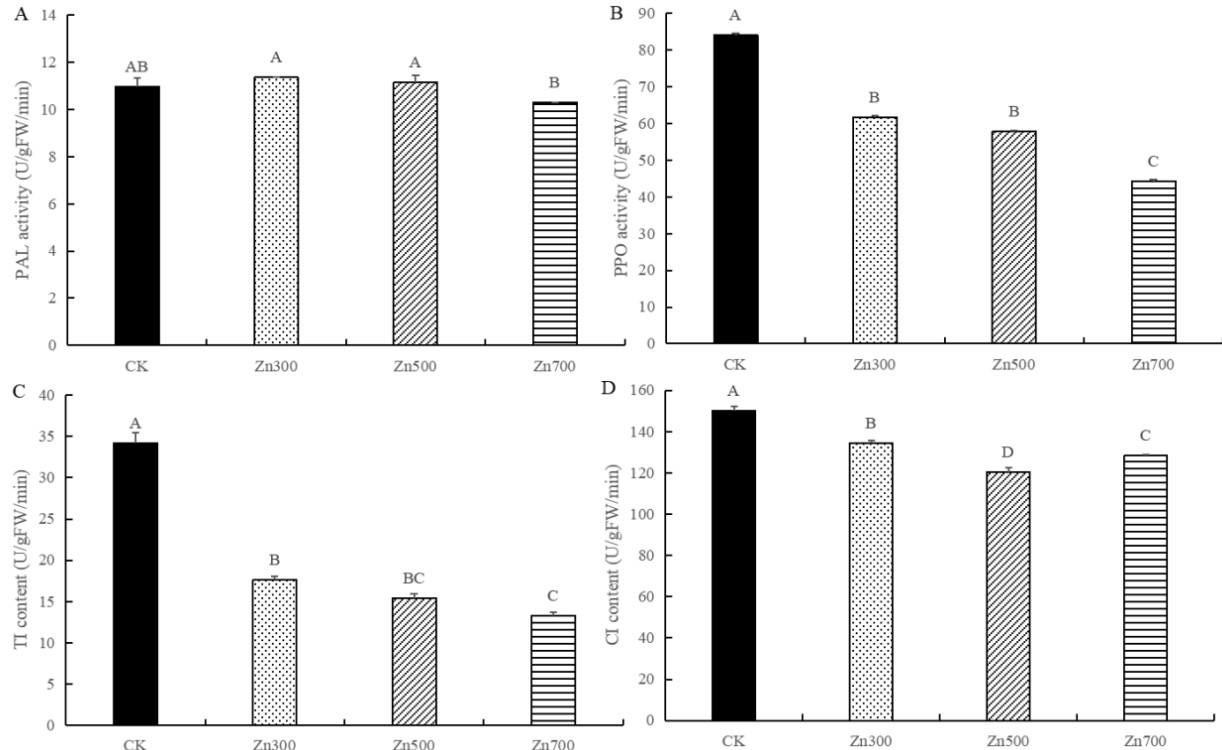

**Figure 3.** Effects of Zn treatment on the activities of PAL (**A**), PPO (**B**), TI (**C**) and CI (**D**) in *P. simonii*. The values shown in the figure are the average value ± SD (n = 3). Different letters indicate that there are significant differences among groups (one-way ANOVA, *p* < 0.05).

*3.5. Effects of Zn Stress on Resistance of P. simonii to Leaf Rust*

The susceptibility of *P. simonii* to pathogenic fungi under Zn stress was measured using the incidence and disease index of *P. simonii* rust under natural conditions. As shown in Figure 4A, Zn stress had no effect on the incidence of *P. simonii* rust. The Zn treatment groups had a significantly higher *P. simonii* rust disease index than the CK group, with that in the $Zn_{700}$ treatment group being significantly higher than those in the $Zn_{300}$ and $Zn_{500}$ treatment groups (Figure 4B). In the $Zn_{300}$, $Zn_{500}$, and $Zn_{700}$ treatment groups, compared with the control group, the rust disease index increased by 29%, 25%, and 42%, respectively.

*3.6. Correlation Analysis between Defense Level and Incidence of P. simonii*

To evaluate whether the element defense and chemical defense levels mediated by heavy metals can affect the disease resistance of *P. simonii*, we analyzed the correlation between the chemical defense index, leaf heavy metal content, and disease resistance correlation index of *P. simonii* (Table 3). The phytochemical parameters CI, TI, and PPO were negatively and significantly correlated with the disease index, while PAL, CAT, and SOD were not significantly correlated with the disease index. There was no significant correlation between all chemical defense parameters and disease incidence. The Zn content in the leaves was positively correlated with the disease index of *P. simonii*, with no obvious correlation with the disease incidence.

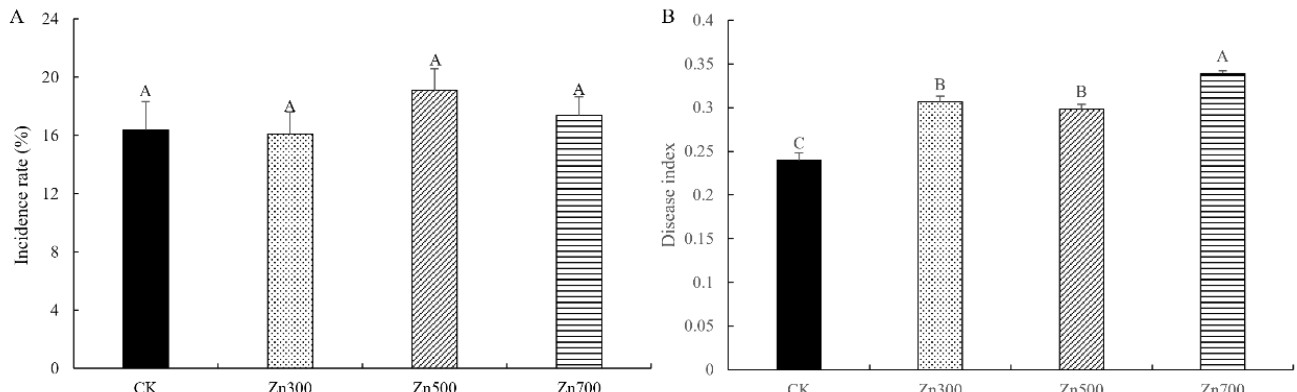

**Figure 4.** Effects of Zn treatment on the incidence rate (**A**) and disease index (**B**) of rust in *P. simonii*. The values shown in the figure are the average value $\pm$ SD (n = 3). Different letters indicate that there are significant differences among groups (one-way ANOVA, $p < 0.05$).

**Table 3.** Correlation analysis between pathogenesis parameters of leaf rust and Zn content or chemical defense parameters.

|  | CAT | SOD | PAL | PPO | TI | CI | Zn Content |
|---|---|---|---|---|---|---|---|
| Disease incidence | −0.01 | −0.05 | −0.32 | −0.19 | −0.18 | −0.31 | 0.26 |
| Disease index | −0.54 | −0.52 | −0.27 | −0.95 * | −0.90 * | −0.75 * | 0.91* |

Note: *, $p < 0.05$ (Z-test).

## 4. Discussion

As a vital part of the food chain, plants may suffer from adverse environmental factors (heavy metals) during the processes of energy metabolism and material exchange with the external environment [34]. These unfavorable factors inhibit their growth and development. For example, Alam et al. (2021) found that heavy metal contamination of the soil could significantly inhibit the growth of tomato plants [35]. In this study, as compared with the CK, Zn treatments with various concentrations reduced the growth and biomass of *P. simonii*. This indicated that even the essential metal elements for plant growth can cause heavy metal toxicity when their content exceeds the tolerance threshold of plants. Similar results were reported by Jiang et al. (2018), who found that the lowest Zn concentration (300 mg/kg) treatment slightly inhibited the growth and development of *Carya illinoinensis*, while the middle (500 mg/kg) and high (700 mg/kg) concentrations of Zn treatment strongly inhibited their growth [30].

In this study, the root length, root dry weight, and root fresh weight of *P. simonii* were inhibited by the different Zn treatments, thereby indicating that *P. simonii* roots have been poisoned by Zn exposure. Additionally, we found that Zn concentration in the roots was significantly lower than that in their aerial parts. With the increase in Zn concentration, the transport coefficient of Zn increased significantly. This indicated that under high Zn stress, *P. simonii* partially mitigated the toxic effects of heavy metals on roots by increasing the transfer of heavy metal from the root to the shoot. As an effective oxidative stress mechanism, the antioxidant defense system comprising antioxidant enzymes was used by plants to alleviate ROS-induced oxidative damage under heavy metal stress [36,37]. In this study, Zn treatment groups significantly inhibited the activities of antioxidant enzymes of *P. simonii*. This indicated that Zn stress impaired the antioxidant defense mechanism of *P. simonii*. Our results were similar to those of Xu et al. (2006), who found that under Zn stress, the ryegrass SOD activity of each Zn treatment group was lower than that of the CK [38]. Interestingly, in this study, the activity of antioxidant enzymes in the high Zn treatment group was significantly higher than that in the low Zn and medium Zn treatment

groups. Therefore, this indicated that *P. simonii* could partially mitigate the toxic effects of high Zn exposure by increasing the activity of the antioxidant enzymes.

Plant defense enzymes, including PAL and PPO, are the key enzymes that synthesize toxic secondary metabolites in plants that are important for resisting biotic stresses (pathogenic bacteria and phytophagous insects) [39,40]. Our study found that as compared with the CK, Zn stress did not significantly affect the PAL activity, but significantly reduced the PPO activity in *P. simonii* leaves with a negative correlation with the Zn concentration. PPO, an important physiological index that represents the plant's chemical defense ability, can increase the resistance of plants to subsequent biotic stresses by converting phenolic compounds into quinones [41]. These results indicated that the synthesis of chemical defense substances or the secondary metabolism of *P. simonii* was inhibited under Zn stress. Therefore, we could infer that Zn stress may increase the sensitivity of *P. simonii* to subsequent biotic stressors (diseases or pests). Protease inhibitors are a class of important plant defense proteins that improve the plant's abiotic and biotic stress resistance abilities [42]. Therefore, protease inhibitors are also an important index of the chemical defense level of plants. In this study, we found that Zn stress significantly inhibited the activities of protease inhibitors (TI and CI). Combined with the decrease in plant defense enzyme activities and the decrease in TI and CI activities, we demonstrated that the overall chemical defense level of *P. simonii* decreased significantly under Zn stress, with increased sensitivity to subsequent pathogens or phytophagous insects.

Multiple studies have shown that heavy metal stress can induce plants to increase their resistance to pathogenic microorganisms [43,44]. For example, Gallego et al. (2017) found that the resistance of *Noccaea caerulescens* to the pathogenic *Alternaria brassicicola* increased under Zn stress and attributed this to the protection conferred by the "elemental defense" [45]. Elemental defense hypothesis proposes that high concentrations of metals, metalloid or non-metallic trace elements can protect the plants against herbivores or pathogenic microorganisms [45,46]. Our heavy metal enrichment analysis showed that Zn enrichment significantly increased Zn concentration in the roots, stems, and leaves, compared to control plants. This indicated that *P. simonii* had a strong Zn translocation ability in a Zn-polluted ecosystem, and this accumulated Zn might endow *P. simonii* with the element defense ability. However, we found that Zn stress increased the susceptibility of *P. simonii* to leaf rust, as evident from the increase in Zn concentration and the significantly increased disease index of *P. simonii* rust. Combined with the determination of phytochemical defense parameters in the Zn treatment group, we deduced that this enhanced sensitivity may be due to the inhibition of the Zn treatment of defense enzymes or protease inhibitors. Therefore, in this study, the Zn-mediated elemental defense did not compensate for the reduced chemical defense in *P. simonii*, thereby weakening the disease resistance of *P. simonii* treated with Zn in the natural environment. Further correlation analysis found that the disease severity of *P. simonii* rust was positively correlated with Zn content in leaves but negatively correlated with the chemical defense parameters of *P. simonii*. These findings clearly supported our above deduction. Of note, we observed the occurrence of *P. simonii* rust in the CK and Zn treatment groups, with the Zn treatment not affecting the incidence of *P. simonii* rust. This indicated that in the natural environment, *P. simonii* has poor resistance to rust, and the risk of its natural infection with rust is high.

## 5. Conclusions

In Zn-contaminated areas, the biotoxic effects of Zn can cause poor plant growth, similar to the non-essential elements. In addition, the adverse effect of Zn on plants is also reflected in the defense level of plants against pathogenic fungi. The accumulation of Zn in plants will increase the pathogenicity of pathogenic fungi to plants. That is, plants living in Zn-contaminated areas may be more susceptible to severe disease outbreaks. This adds an abiotic factor to the prevention and control of tree diseases in heavy metal-polluted areas. Since our study on plant disease susceptibility was a completely natural process, it was impossible to ensure that the initial infected spores of leaf rust were evenly

distributed among the different plants. In our follow-up studies, we will accurately identify the influence of heavy metal pollution on plant disease susceptibility through artificial inoculation.

**Supplementary Materials:** The following supporting information can be downloaded at: https://www.mdpi.com/article/10.3390/f14040783/s1, Figure S1: Diagram of seedling cultivation site; Table S1: Physical and chemical properties in soil; Table S2: The enrichment coefficient and transport coefficient of CK and Zn treatment groups.

**Author Contributions:** Methodology, L.G.; Investigation, L.G. validation, L.G. and S.Y.; data curation, L.G.; writing—original draft preparation, A.Z. and L.G.; writing—review and editing, A.Z. and S.Y.; Funding acquisition, S.Y. All authors have read and agreed to the published version of the manuscript.

**Funding:** This research was supported by the Fundamental Research Funds for the Central Universities under Grant 2572020DP15.

**Institutional Review Board Statement:** Not applicable.

**Informed Consent Statement:** Not applicable.

**Data Availability Statement:** All the data that support the findings of this study are available in the manuscript.

**Conflicts of Interest:** The authors declare that they have no competing interests.

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
