# Peer review of "Effects of Zn Exposure on Populus simonii Seedling Growth and Its Resistance to Leaf Rust"

_forests, doi:10.3390/f14040783_

Round 1

Reviewer 1 Report (New Reviewer)

The manuscript deals with the study of effect of differents doses of Zn and leaf rust on accumulation of Zn in different parts of Populus simonii, and on enzymatic activities.

The topic of the manuscript is interesting, the meausurements done are current.

Nevertheless, there are many shortcomings in this manuscript.

1- the experimental design is not presented.

2- there are two studied factors. Authors did not present any interaction between the two studied factors. This is not presented in M&M section. which tends to say that the possible experimental design chosen by the authors would be random blocks. However, if there are two factors being studied, it is preferable to choose a split plot design. AND to do the statistical analyses accordingly.

Therefore, the manuscript needs deep modifications.

further remark

Figure 1

Use the same ordinate scale for the four parts of the figure. This will allow comparison between the different parts of the plant and the soil.

Author Response

Reviewer 2 Report (New Reviewer)

Please see the review comments document.

Round 2

Reviewer 1 Report (New Reviewer)

Authors have considered all requested modifications. Nevertheless, the manuscript needs further minor modifications. Indeed, there are two studied factors. interactions between the two factors are not presented.

Author Response

Thank you very much for your comments on this manuscript. Our CK group and all the Zn-treated groups were infected with rust. This is a naturally occurring phenomenon. As a result, we have no CK and Zn-treated groups that are not infected with rust. Therefore, it is not possible to analyze the interaction between Zn treatment and rust infection. Your opinion is very valuable. When we simulate the occurrence of rust disease through artificial inoculation, the interaction analysis of the two factors will be carried out. This deficiency does not affect the quality of the text. Our correlation analysis has shown that the decrease of chemical defense of seedlings under Zn exposure is the main reason for the increase of rust disease index.

This manuscript is a resubmission of an earlier submission. The following is a list of the peer review reports and author responses from that submission.

Round 1

Reviewer 1 Report

This manuscript titled “Effects of Zn exposure on Populus simonii seedling growth and its resistance to leaf rust” presents essential new data about effects of Zn exposure. This is an intriguing manuscript that presents the results of a well-planned and executed study. There are several shortcomings for that should be resolve.

Line 50: " in vivo " should be " in vivo "

Line 58-65: ".Additionally" should be ". Additionally". In addition, rephrase the sentences with scientific manner.

Line 105, 106, 112: " ºC" should be " °C".

Line 121-122: It will be attractable if the formula presents in separate sentence as equation and inside the paragraph. (Also in Line 202-204, Line 206-209.)

Additional comments: Please find some recent papers in the introduction that discuss heavy metals. Also, include an adequate number of references. Furthermore, discuss the future perspectives and limitations.

Reviewer 2 Report

Dear authors,

The publication is a valuable contribution to the field. The text must be polished, there are some spelling errors.

Line 50: Please use italics “in vivo

Figure 4.B.: Please correct the name of y-axis name, disease is in lower case

Line 106: a graphite digester

Line 107: was completed

Homogenize the nomenclature of units: U/GFW or U/g

I recommend expressing the control in the same way in text and in figures.

Please kindly provide the following data.

Line 103: Under what conditions was the sample stored?

In all the figures you express "Different letters indicate that there are significant differences between the two groups", were the statistics performed each treatment vs. control?

Materials and methods section (Determination of heavy metal content in P. simonii): Please define what concentrations of the reagents you used.

The conclusion must be improved, adding the impact of the accumulated evidence in terms of yield and possible applications.

Best regards,

Reviewer 3 Report

Although the manuscript’s topic is interesting, there are several issues that should be correct before to publish this work.

Line 82. Elimiate “soil”

Line 84. Eliminate “comprised sandy soil, peat soil, and local soil (V:V:V=1: 1: 1). The properties of soil substrate”.

Line 89. How was ZnCl2 applied to the substrate?

Line 93. Were seedlings planted again after treatments application to the substrate? Clarify it.

Table S2. “The values shown in the figure re…” It is not a figure, it is a table.

Authors should indicate the mean comparison test used to analyze data of Table 2 in the legend.

Table S2 “Different letters indicate that there are significant differences between the two groups.” But, which are these two groups?

I do not understand why there are 150 plants in each treatment (line 93), but only 9 seedlings (3 seedlings per 3 replicates) were analyzed. It is a very poor representation of the experiment. The same situation occurs in section 2.3, only 5 plants were evaluated per treatment.

Line 95. It is hard to observe in Fig. S1 what authors described in lines 93 to 95.

 Line 95. How many leaves constitute a sample?

The Figure S1 does not represent a diagram f seedling cultivation site, it is just a photograph.

Line 197. “At the end of August” we do not have more references about time, so we do not know when the experiment started.

Line 197. Was the rust infection a natural process?

Authors called control as CK, however in Figure 1 control is called Zn0. Homogenize the nomenclature to facilitate understanding.

Lines 239, 251, 283. Which two groups?

Table S2. If comparisons are for each column, revise the transfer coefficient column, because letters are wrong.

Line 256. Did authors include Zn0 to make the correlation with CAT and SOD variables?

How can authors explain that the CAT and SOD activity increases with higher Zn concentration, but the maximum values of both variables, CAT and SOD, were registered in control plants?

Line 285. Populus leaf rust is not caused by a bacterium. It is caused by the FUNGI Melampsora spp. Authors should know the disease perfectly before doing and publishing this work.

Section 3.5. As the infection process was natural, we do not know if the inoculum was homogeneously spread among the different plants or not, leading different disease index due to the different treatments or to the pathogen dynamics.

Line 320 and 321. Which are low, middle, and high concentrations of Zn according with Kulbat-Warycha et al. (2021)?

Line 327 and 328. “P. simonii partially mitigated the toxic effects of heavy metals by reducing the contact area between the roots and heavy metals”

Line 330. In their aerial parts instead of its

Line 221. Transport coefficient of Zn

Lines 333 to 336. Include a reference

Lines 336 and 337. This sentence describes results found in this study, so the references 31 and 32 do not make sense in this site.

Line 339 to 341. But this is not true for the present study, because the higher CAT and SOD activities were registered in control plants with 0 Zn.